# End-to-end reproducible AI pipelines in radiology using the cloud

Dennis Bontempi [1,2,3], Leonard Nuernberg[1,2,3], Suraj Pai [1,2,3], Deepa Krishnaswamy [4], Vamsi Thiriveedhi[4], Ahmed Hosny[1,3], Raymond H. Mak [1,3], Keyvan Farahani[5], Ron Kikinis [4], Andrey Fedorov[4] & Hugo J. W. L. Aerts [1,2,3] ✉

Artificial intelligence (AI) algorithms hold the potential to revolutionize radiology. However, a significant portion of the published literature lacks transparency and reproducibility, which hampers sustained progress toward clinical translation. Although several reporting guidelines have been proposed, identifying practical means to address these issues remains challenging. Here, we show the potential of cloud-based infrastructure for implementing and sharing transparent and reproducible AI-based radiology pipelines. We demonstrate end-to-end reproducibility from retrieving cloud-hosted data, through data pre-processing, deep learning inference, and post-processing, to the analysis and reporting of the final results. We successfully implement two distinct use cases, starting from recent literature on AI-based biomarkers for cancer imaging. Using cloud-hosted data and computing, we confirm the findings of these studies and extend the validation to previously unseen data for one of the use cases. Furthermore, we provide the community with transparent and easy-to-extend examples of pipelines impactful for the broader oncology field. Our approach demonstrates the potential of cloud resources for implementing, sharing, and using reproducible and transparent AI pipelines, which can accelerate the translation into clinical solutions.

Recent advances in artificial intelligence (AI) are helping to address many challenges in different medical areas, such as radiology, radiation oncology, and pathology[1–4]. Several factors contributed to these notable AI breakthroughs, including the progressive increase in computational processing power driven by the development of specialized hardware, the recent advances in deep-learning (DL) algorithms, the availability of widely used open-source DL platforms, and the ever-growing amount of publicly available data to train and validate AI models on. Nonetheless, the lack of transparency and reproducibility of published research, as well as processing pipelines, remains one of the biggest challenges faced by the field[5]. Recent studies[6,7] suggest that

as little as one-quarter of the AI publications are accompanied by all the resources required to replicate the findings as reported. Although this so-called reproducibility crisis is not exclusive to AI[8], the sheer complexity and black-box nature of DL algorithms, the large amount of data required to develop and validate models with millions of parameters, and the poor documentation and sharing of AI research[5] substantially worsen the problem.

Transparency and reproducibility are of utmost importance, particularly in the medical field, where errors may result in unpredictable and potentially untoward consequences. While AI can improve the efficiency and effectiveness of patient care[9], poor

[1]Artificial Intelligence in Medicine (AIM) Program, Mass General Brigham, Harvard Medical School, Boston, MA, USA. [2]Radiology and Nuclear Medicine, CARIM & GROW, Maastricht University, Maastricht, The Netherlands. [3]Department of Radiation Oncology, Dana-Farber Cancer Institute, Harvard Medical School, Boston, MA, USA. [4]Department of Radiology, Brigham and Women's Hospital, Harvard Medical School, Boston, MA, USA. [5]National Heart, Lung, and Blood Institute, National Institutes of Health, Bethesda, MD, USA. ✉e-mail: haerts@bwh.harvard.edu

implementations can introduce significant biases, exacerbate health disparities[10], and even lead to misdiagnoses and suboptimal treatments[11,12]. Inadequate efforts in making AI research reproducible make performance claims very difficult to verify[13], ultimately leading to inflated accuracy rates[14] and generalizability problems[15,16], hindering the translation of these systems to the clinic[17]. Furthermore, such a lack of reproducibility often impedes comparing performance across different AI tools.

Several reporting guidelines, checklists, and standards have been proposed in recent years to tackle these issues in the medical imaging field[18–24]. These protocols aim to instruct scientists on the best practices in documenting their research and help them identify aspects of the AI-based analyses that should be reported in publications to ensure reproducibility. Moreover, community-driven efforts led to the creation of other resources, such as a registry of AI models developed for biomedical applications[25] that researchers can use to report minimal attributes of these systems. While these efforts present authors with the rules to follow during the experimental design process and manuscript preparation, they do not provide a practical way to share AI-based pipelines with the community. Even if part of the code is shared, building the pipeline to reproduce the results reported in a study is often left as an exercise for the reader[26] and, as thoroughly investigated, this can substantially change the result of computational pipelines when ill-executed[5,27–29]. Furthermore, these pipelines often need domain-specific tools, specific software versions, or hardware capability to be reproduced—or even run—as intended without encountering errors, with this information being regularly not provided to the reader. Without the accurate implementation of all the steps, any community attempt to expand on a study or to identify critical issues with (and failing conditions for) a pipeline remains practically impossible. Overall, in contrast to other research fields being transformed by technologies—e.g., genomics, where the MIAME guidelines[30] and efforts like the NCI Genomics Data Commons have been instrumental in improving reproducibility by establishing standardized reporting and facilitating data sharing[31,32]—the lack of reproducibility in AI research for medical imaging is significantly more widespread[33].

Cloud-based resources can facilitate a solution to these issues and improve the reproducibility and scalability of AI studies by revolutionizing the way researchers access and use data in their studies[34]. The use of cloud platforms can facilitate access to large-scale cancer imaging data while delivering a more streamlined and consistent computing environment to conduct experiments. Furthermore, by providing public access to data, software, and hardware, cloud-based resources can enable researchers to inspect all the components needed to replicate AI studies, ultimately contributing to the growth and advancement of the field.

In this study, we used a cloud-based infrastructure for the implementation of end-to-end reproducible and transparent AI pipelines (see Fig. 1). By leveraging the NCI Imaging Data Commons (IDC)[35] for data and the Google Cloud Platform (GCP) for computing, each step within the AI pipeline—from data retrieval and preprocessing, deep-learning inference, data post-processing, and the analysis of the final results—could be examined and reproduced. We show end-to-end replication of two published AI studies. The first use case is the implementation of a deep-learning model that enables outcome prediction in patients with non-small cell lung cancer (NSCLC)[36], where we were able to extend the study on a previously unseen portion of the validation dataset. The second use case describes the implementation of a novel foundation model for the discovery of quantitative imaging biomarkers with potential for the broader field[37]. We replicated these studies and documented all the steps to do so, providing the community with working examples of how to extend these studies to new datasets. We share notebooks for the visualization, exploration, and interpretation of the results so users can replicate the findings. As a result, the developed end-to-end analysis pipelines complement the published manuscripts and the accompanying source code repositories, enabling the user to easily experience the benefits of the proposed approach. Our workflow demonstrates the potential of cloud-based resources for implementing, sharing, and using reproducible and transparent AI pipelines, which can accelerate the translation of AI algorithms into the clinic.

## Results

### Use Case I: deep learning for lung cancer prognostication

For the first use case, we implemented a convolutional neural network (CNN) for lung cancer patient stratification[36]. In the original publication, the authors used both clinical and imaging data (standard-of-care computed tomography (CT) scans) of NSCLC patients to assess the performance of the deep-learning pipeline. Although the authors shared an open-source implementation of the algorithm, the model weights, the code to run inference, and a written description of how to prepare data for inference, the data preprocessing component was not included. One of the independent testing cohorts used in this study is now hosted by the IDC (i.e., the NSCLC-Radiomics dataset[38]), allowing us to use the cloud platform to reproduce the AI pipeline end-to-end. We provide an overview of the re-implementation in Fig. 1b.

In order to replicate the tumor lesion analysis, we used expert segmentations of the 3D tumor volumes included in the NSCLC-Radiomics collection, also hosted by the IDC. These masks are a newer iteration of the set of labels used for the original publication. As reported in Fig. 2, the AUC of the original pipeline, computed on the cohort of $N = 206$ patients, was AUC = 0.7, while the replicated pipeline scored AUC = 0.68 when run on the same subset. After testing with the two-sided Mann–Whitney $U$ test and the DeLong test for paired AUC curves, we observed that the difference between the results produced by the pipeline originally published by Hosny et al.[36] and those generated by our pipeline are not statistically significant (two-sided Mann–Whitney $U$ test $P > 0.05$, $N = 206$; DeLong test for correlated (paired) AUC curves $P > 0.05$, $N = 206$).

Furthermore, we conducted a Kaplan–Meier analysis to assess the stratification power of the AI pipeline. We found that both the original pipeline and the replicated pipeline can successfully stratify higher-risk patients from lower-risk patients ($P < 0.001$ and $P = 0.023$, for the original and the replicated pipeline, respectively; see Fig. 2) when the risk-score threshold shared with the original publication is used to compute the split. When a Cox proportional hazards (PH) model was fitted, we found that the high-risk group had a hazard ratio (HR) of 1.45 (95% CI 1.05–2.00, $P < 0.05$), compared to the low-risk (baseline) group on the subset of the NSCLC-Radiomics dataset the authors used. This was similar to the results reported in the original publication (HR of 1.75, 95% CI 1.26–2.44, $P < 0.005$).

Finally, we extended the analysis to the subset of patients that was excluded from the original publication due to missing clinical and or imaging data in a previous iteration of the dataset, testing the pipeline on $N = 421$ patients (after excluding one patient without a tumor segmentation mask). On the whole NSCLC-Radiomics dataset, the pipeline performance dropped, even though the model retained prognostic power (AUC = 0.61, $N = 421$). The high-risk group showed a hazard ratio (HR) of 1.31 (95% CI 1.04–1.65, $P < 0.05$) compared to the low-risk (baseline) group on the whole dataset. These results indicate that the cloud-based model implementation was successful, that the model performance remained significant on a previously unseen portion of the validation dataset, and that the model was robust to variation in the input segmentation mask, as the original work claims.

### Use Case II: foundation models for quantitative biomarker discovery

For the second use case, we selected a recent study highlighting the potential of foundation models for quantitative biomarker discovery

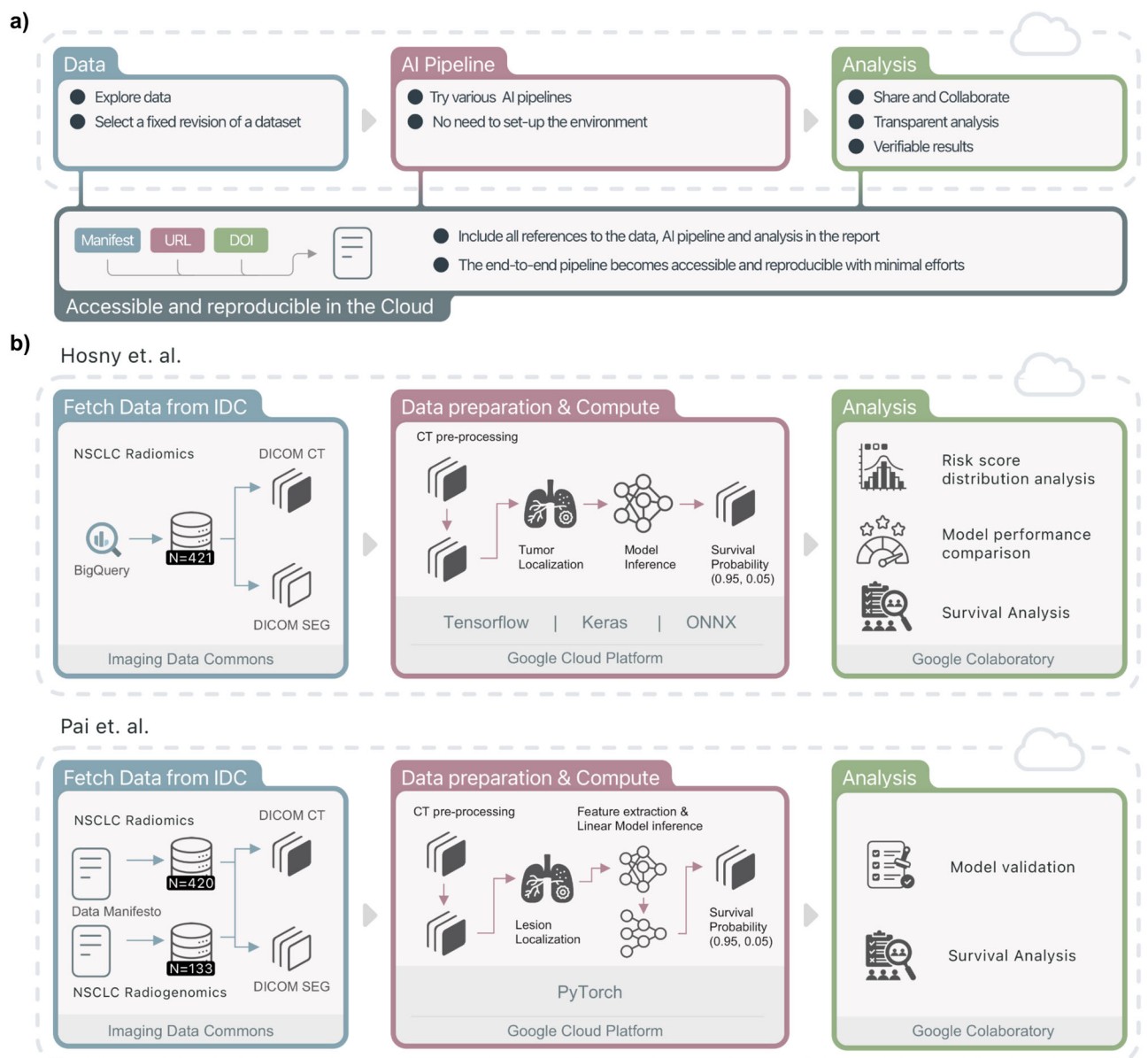

**Fig. 1 | Study Overview. a** The proposed approach combines open-source components, commercially available tools, and infrastructure being established by the US National Cancer Institute to implement reproducible image analysis workflows. **b** We implemented two cloud-based pipelines from the AI in radiology literature (Hosny et al. for Use Case I and Pai et al. for Use Case II), showcasing how the proposed workflow can be used to validate and extend previous studies. For both use cases, we used cloud-hosted data cohorts from the Imaging Data Commons (IDC), and components from the Google Cloud Platform to provision pre-configured GPU-enabled resources for the execution of AI pipelines and their evaluation, including performance, statistical, and survival analysis. The combination of these different tools ensures easily accessible end-to-end pipelines that can be run, inspected, and easily referenced in all of their components. Credit (icons): FlatIcon.

in cancer imaging[37]. This study presents a foundation model that can be used for several applications in quantitative cancer imaging. The authors used self-supervised learning to train a foundation model with strong generalization performance across diverse use cases (lesion's anatomical site prediction, nodule malignancy, and tumor prognosis) and cancer datasets. We provide an overview of the re-implementation in Fig. 1b. We used the IDC platform to define a data cohort matching the subset of the NSCLC-Radiomics and NSCLC-Radiogenomics datasets used in the original publication. Using the data manifest extracted from the IDC platform (which we make publicly available within the notebooks), we retrieved the necessary imaging data for the cohort in a transparent and reproducible manner.

Using the tools provided by the authors, the expert segmentations included in the NSCLC-Radiomics and NSCLC-Radiogenomics

collection on IDC, and the clinical data available for both collections, we successfully replicated the ROC and the KM analyses in the original publication, with the linear model built from the foundation features achieving an AUC of 0.64 on the NSCLC-Radiomics (LUNG1) collection ($N = 420$, after excluding patients with missing clinical or survival data) and of 0.65 on the NSCLC-Radiogenomics collection ($N = 133$, after excluding patients with missing clinical or survival data), and showing good stratification performance in the KM analysis ($P < 0.001$ and $P = 0.006$ on the two cohorts, respectively; see Fig. 3). Using Cox proportional hazards analysis, we found that the high-risk group had a hazard ratio (HR) of 1.43 (95% CI 1.16–1.76, $P < 0.005$) compared to the low-risk (baseline) group on the NSCLC-Radiomics dataset. On the NSCLC-Radiogenomics dataset, the HR for the high-risk group was 2.4 (95% CI 1.29–4.46, $P < 0.01$), consistent with the results reported in the

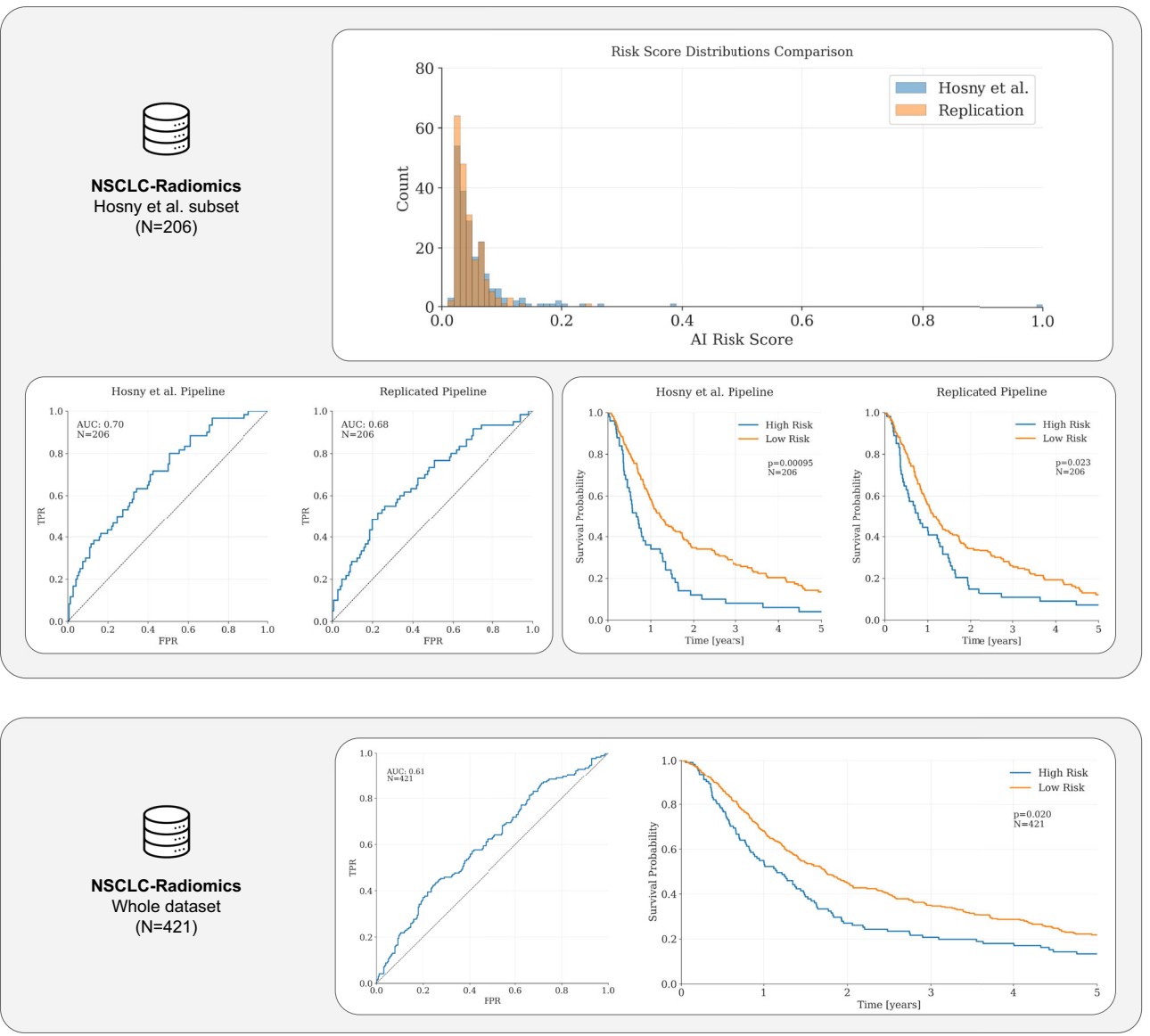

**Fig. 2 | Results for Use Case I–Hosny et al.** We validated Hosny et al. on the non-small cell lung cancer (NSCLC) Radiomics dataset. We compared the cloud implementation to the original publication on the subset of data from the NSCLC-Radiomics dataset the authors originally used. Despite the imaging cohort being updated over time, we found no statistically significant with the original results (AUC = 0.7 and AUC = 0.68 for the original pipeline and the cloud-based re-implementation, respectively; DeLong test for paired AUC curves $P = 0.62$, $N = 206$). The difference between the risk-score distributions was not statistically significant (two-sided Mann–Whitney $U$ test $P > 0.05$, $N = 206$). Both models showed a good stratification power (Hosny et al. HR = 1.75, 95% CI 1.26–2.44, $P < 0.005$; cloud-re-implementation HR = 1.45, 95% CI 1.05–2.00, $P = 0.02$). Furthermore, we extended the analysis to a subset of patients that were excluded from the original publication due to missing clinical and or imaging data in a previous iteration of the dataset. We found that, even if the pipeline performance dropped significantly, the model retained some of its prognostic power (AUC = 0.61, $N = 421$; HR = 1.31, 95% CI 1.04–1.65, $P = 0.02$). Credit (icons): FlatIcon.

paper. These results demonstrate the successful implementation of the foundation model in cloud-based platforms, as the model demonstrated similar performance on the cloud-based data in IDC.

## Discussion

In this paper, we showcased the potential of cloud-based infrastructures for implementing and sharing transparent and reproducible AI-based radiology pipelines. We demonstrated how cloud-based resources offer potential solutions to issues surrounding the reproducibility of AI studies. By providing a consistent computing environment, simplifying data exploration and access, and enabling the storage and sharing of code and results, these resources facilitate the implementation and sharing of fully reproducible AI pipelines that complement the original publications. We demonstrated the effectiveness of our proposed workflow in replicating and disseminating published AI literature. In addition, this workflow offers researchers and practitioners a practical approach to operating AI pipelines while simultaneously promoting best practices and open software for AI in radiology, thus contributing to a more harmonized environment for computational techniques.

The Imaging Data Commons (IDC)[35] provides several capabilities that are key to enabling this study. First, the IDC provides persistent and highly efficient access to cancer research data by hosting it in cloud buckets that are accessible via an open, high-performance interface. Further, the data are harmonized and standardized using the DICOM standard, enabling interoperability and reuse of various tools implementing the standard. Second, besides centralizing and providing easy access to a wide range of standardized medical imaging data,

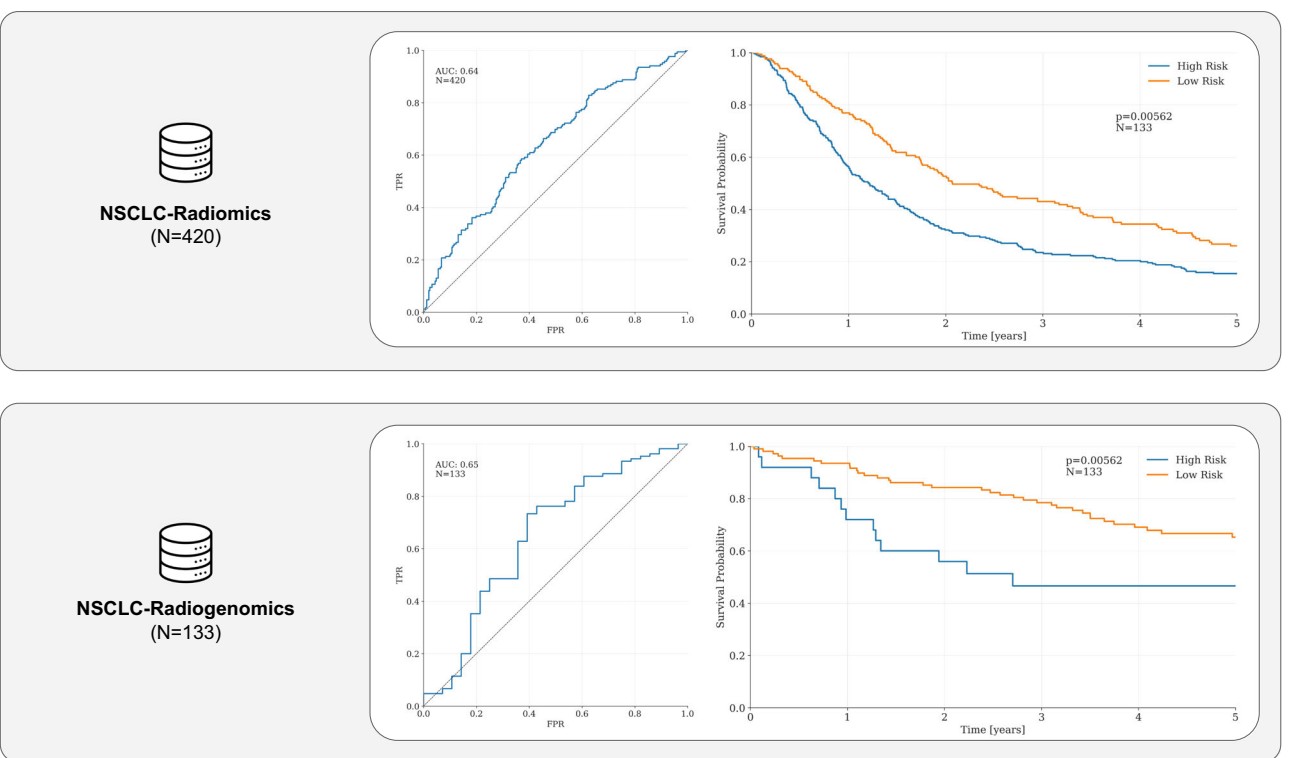

**Fig. 3 | Results for Use Case II—Pai et al.** We replicated the findings of Pai et al. on the non-small cell lung cancer (NSCLC) Radiomics dataset and the NSCLC-Radiogenomics dataset. The linear model built on top of Pai et al.'s foundation model achieved an AUC of 0.64 on the NSCLC-Radiomics collection ($N = 420$) and of 0.65 on the NSCLC-Radiogenomics collection ($N = 133$) and significantly stratifying patients in the Kaplan–Meier analysis ($P < 0.001$ and $P = 0.006$ on the two

cohorts, respectively). The high-risk group had a hazard ratio (HR) of 1.43 (95% CI 1.16–1.76, $P < 0.005$) compared to the low-risk (baseline) group on the NSCLC-Radiomics dataset, while on the NSCLC-Radiogenomics dataset, the HR for the high-risk group was 2.4 (95% CI 1.29–4.46, $P = 0.01$), consistent with the results reported in the original publication. Credit (icons): FlatIcon.

the IDC allows users to create cohorts of imaging study and image-derived data based on DICOM metadata, thus effectively enabling the validation of medical imaging use cases using public datasets. Finally, since the platform supports data versioning, the specific cohorts utilized in our study will remain accessible in IDC without requiring any maintenance from the data submitters. This ensures long-term preservation and accessibility of imaging data for reproducibility and future research while helping to associate a set of processing results with a specific cohort version. As a data commons, IDC provides a range of tools that aim to support the end-to-end process of algorithm development and continuous data enrichment, including hosted viewers to enable visualization of both the images and image-derived data, tools for conversion of the AI analysis results to standard DICOM representation and procedures for sharing the generated analysis results as a complement to the analyzed images.

We decided to use Google Colaboratory as a cloud-based computational resource as the platform provides users with a Jupyter Notebook environment for writing and running Python code alongside markdown annotation from a web browser—eliminating the need to set up a local runtime environment—and enables us to easily share minimal working examples and full-fledged analyses. To bolster transparency and reproducibility, the user can integrate text, visualizations, and code in the notebooks part of this resource, facilitating a comprehensive communication of the tools we used, the inner workings of the data preparation and processing pipeline, and the research findings.

As for the use cases, there are several reasons we chose Hosny et al. and Pai et al. for this study. First, since both models were originally developed and validated by our research group, we had a deep understanding of the models and all of the steps necessary for

replication—a necessary condition in order to limit imprecisions and flaws in the implementation of cloud-based resources for an AI pipeline. Second, we believe both studies are relevant in the field of deep learning in radiology. Hosny et al. is one of the field's first and most cited studies developing deep radiomics techniques for patient prognosis, while we believe the foundation model from ref. 37 has potential for broad applications in cancer research and could benefit from the dissemination of a cloud-based resource that the community can use to verify, in a transparent and reproducible manner, the performance the model. Finally, both Hosny et al. and Pai et al. validated the performance of their models on two non-small cell lung cancer datasets hosted by the IDC, i.e., the NSCLC-Radiomics dataset and the NSCLC-Radiogenomics—allowing us to use the platform for the development of our cloud-based workflow.

While cloud-based infrastructure facilitates potential solutions to issues surrounding the reproducibility of AI studies, we recognize its adoption also comes with several challenges, the biggest of which is the learning curve for the user unaccustomed to the different interacting blocks and tools. Such a learning curve can vary depending on a researcher's background and familiarity with cloud computing, and it typically involves gaining an understanding of cloud infrastructure and services and learning how to use cloud-based tools for data storage, processing, and analysis. However, many cloud providers and cloud resources (such as Google Colaboratory, GCP, the CRDC, and IDC) offer comprehensive documentation, training material, and access to a growing community of practitioners to provide guidance and support. Furthermore, this process could be substantially facilitated by the rapid expansion of the cloud-computing landscape, with many alternatives being used and promoted in medical imaging and in the AI field nowadays. A few notable examples besides the Google Cloud Platform

are Amazon Sagemaker Studio Lab[39] (used by platforms such as Grand-Challenge[40] and Hugging Face[41]) and Microsoft Azure ML (also supported by Hugging Face[41]). Several other cloud-based platforms, such as Binder[42] and Notable[43], are blooming—but might not be ideal for medical imaging or GPU-intensive pipelines (as the ones presented in this work).

The cloud offers a significant advantage for researchers, especially those lacking access to in-house data, who cannot generate data at the same scale as what is available from environments such as IDC or who lack the infrastructure for storing and processing large datasets. However, it must be recognized that using the cloud for data sharing, particularly for reproducibility purposes, involves additional steps. For instance, anonymization and institutional approval are often necessary to address security and privacy concerns about sharing protected health information, such as medical images and associated clinical data. In the cases when, for the aforementioned reasons, sharing data is impractical, the users might prefer to opt for techniques such as federated learning, as it enables collaborators to build and refine AI models without sharing raw data (only model updates are exchanged, ensuring that sensitive patient information remains localized and secure). Furthermore, the proposed solution assumes that the data needed for the notebook and the cloud-computing platform itself are available in the region in which the user is trying to reproduce the experiments. Governance of the data access and the cloud-computing access is outside of the scope of this manuscript, but it is important to recognize this can pose a limitation to our workflow. Nevertheless, we believe the cloud can be of great help in overcoming system limitations often encountered in large-scale research studies.

Cloud computing offers a range of advantages over on-premises computation, but it does not eliminate the utility of the latter. Rather, they offer an alternative that should be recognized by the research community. One of the benefits of cloud computing is the easy and ubiquitous access to a computational environment that requires no experience to be set up and maintained, which researchers can leverage without upfront investments in hardware or infrastructure. The cloud offers extreme scalability and elasticity of the available resources and allows one to easily access the latest computational capabilities without investing in costly upgrades. While this can be particularly advantageous for small research teams or individuals who may not have access to powerful computing resources, using cloud computing can lead to dependence on third-party providers, which can be a concern for researchers who may need to access resources and data in the long term. Cloud-computing providers may change their pricing models or even discontinue services, which could disrupt the research workflow. In addition, cloud computing may limit the control and flexibility of the user in some regards. For example, researchers may be limited in using obsolete code that requires specific hardware or software—although this problem is not limited to cloud resources but can easily extend to local computational clusters. Furthermore, cloud providers might update these components without appropriately informing the users, resulting in the disruption, or even failure, of the previously developed pipelines. Even if this does not entirely hinder the benefits of the added transparency, it is essential to note the implementation and sharing of resources following the proposed workflow and through the aforementioned cloud services do not eliminate the need to maintain the code repository over time. Finally, cloud computing may not always be the most cost-efficient solution for all use cases, particularly for long-term or high-usage applications.

Although we strongly believe using the cloud can bolster transparency and reproducibility in the field of AI for radiology, we recognize that, in some cases, there might be other factors at play that limit such good practices in our field. For instance, we acknowledge that Intellectual Property Rights (IPR) can pose challenges, especially when such AI models are being validated (or planning to be validated)

further in a clinical setting (e.g., in perspective clinical trials) or when they can be of commercial value for companies funding the research—as a significant part of AI research in radiology, and, more generally, medical imaging, is nowadays driven by industry. Moreover, in such a highly competitive field, researchers are often caught in a race against time, striving for rapid development and deployment to maintain a competitive edge. This urgency can lead to a reluctance to fully embrace transparency, as research groups may fear that sharing too much information could have a negative impact on their advantage. However, the proposed workflow is not aimed at researchers who are bound by the aforementioned constraints. Rather, we see the workflow as a step toward the solution for researchers who are willing to accompany published literature with resources that enable independent validation and who don't see the current guidelines and checklists as a pragmatic way to do so. To this day, reproducing AI studies in the field of radiology (and, more generally, in healthcare), as widely documented in the literature, remains a challenge even for the more technically proficient users. Importantly, in such a competitive field, researchers, and also—perhaps, more importantly—providers and patients are also caught in a race against time to gain access to the latest advances in AI in order to evaluate the robustness of the latest tools and ultimately expedite translation of the robust algorithms into the clinic. Although we recognize there is space for improvements, we believe the proposed approach may help enable such expedited translation by simplifying the process for developers and users.

In summary, cloud computing offers several advantages for reproducibility in AI research, including democratizing access to large standardized datasets and free-to-use, easily accessible, consistent computational environments. There still are essential factors to consider when using cloud computing in AI research, often cost-related—but we believe embracing cloud technology could reduce its costs and ultimately yield the providers of cloud-computing resources to develop more customized and cost-effective solutions for the medical image analysis research community. Finally, the long-term cost savings of cloud adoption span beyond direct infrastructure and computing expenses. By enabling greater collaboration and data sharing, cloud technology can accelerate medical AI research and the development of more accurate, effective, and personalized clinical tools. This underscores the compelling case for embracing cloud technology in medical AI research.

## Methods

### Data from Imaging Data Commons

We retrieved two collections with imaging data from the Imaging Data Commons (IDC)[35,44], a cloud-based environment for publicly available cancer imaging data integrated with analysis and exploration tools. The first is the NSCLC-Radiomics collection[38,45] (also known as LUNG1 dataset), consisting of 422 patients with stage I–IIIb non-small cell lung cancer, treated with radiotherapy or chemo-radiation therapy at the MAASTRO Clinic in The Netherlands, all of which were imaged with CT, with or without intravenous contrast. For all of the patients, a manual segmentation of the tumor was drawn by a radiation oncologist for treatment purposes. We used the IDC platform to define a data cohort matching the subset of the NSCLC-Radiomics dataset used in the original publication. By querying the accompanying metadata curated in IDC tables using SQL, we retrieved the necessary imaging data for the cohort in a transparent and reproducible manner. The query to generate the imaging data cohort (and its results) are publicly available within the notebooks (and in Supplementary Item 1), together with a description of the procedure used to select the validation cohorts for our study. The resources we share also describe the details and provide an example of how to run the queries to generate the validation cohort.

The second dataset is the NSCLC-Radiogenomics collection[46,47]. This dataset includes CT images for 211 NSCLC stage I–IV patients acquired at the Stanford University School of Medicine and at the Palo

Alto Veterans Affairs Healthcare System. The dataset also features segmentation maps of the tumors, manually drawn from the CT scan by several radiation oncologists, and semantic annotations of the tumors as observed on the medical images using a uniform vocabulary. In addition, the imaging information is matched with clinical and survival data. The query to generate the imaging data cohort is available in Supplementary Item 2, while in the notebooks, we provide a data manifest to do so.

## Cloud-based implementation of DL pipelines

We implemented our cloud-based AI pipelines for both use cases using the computing infrastructure provided by Google Colaboratory[48] (or Colab), sharing additional resources researchers can use to replicate and build upon the study. The Colab Notebooks were built on top of the standard Colab environment and were tested using several Colab configurations (i.e., "standard" CPU-only, "high RAM" CPU-only, "free" GPU, and "pro" GPU). Additional details are available in Supplementary Table 1. While defining such a computational environment for medical image processing, we selected tools that are open-source and actively developed (or maintained). For most of the medical image preparation and DICOM CT data preprocessing tasks, in both use cases, we used Plastimatch[49], Numpy[50], and the Python APIs to SimpleITK[51] (which is a simplified, open-source interface to the Insight Segmentation and Registration Toolkit[52]). To provide visual insights into the segmentation masks and 3D medical images in the notebooks, we used ITKWidgets[53], a utility that provides interactive widgets to visualize images, point sets, and meshes in 3D or 2D. We also provided, whenever possible, a link to the Imaging Data Commons viewer for radiological images, a browser-based zero-footprint DICOM viewer based on the OHIF[54] viewer. To cross-load cloud-hosted data to the Colab instances, we used s5cmd[55], a platform-agnostic open-source alternative to the tools developed by cloud providers. Since this data cross-loaded from the Imaging Data Commons is saved in DICOM format, we used dicomsort[56], a tool that provides custom sorting and renaming of DICOM files to uniform their structure before processing. Other tools we used in various operations involving the reading, conversion, or preparation of DICOM files are pydicom[57] and dcmrtstruct2nii[58]. For the evaluation metrics and for survival analysis, we used Scipy[59], Scikit-learn[60], and Lifelines[61]. All of the packages' and tools' versions are available as part of our GitHub project repository and can be retrieved directly from the Colab notebooks. To make it easier for others to extend both studies, we share notebooks incorporating comprehensive descriptions of every step of the replication process to help users navigate the complexity of the pipelines. Finally, we share several notebooks with the code used to generate the results we present in this publication and the artifacts resulting from both pipelines (i.e., the risk scores for the prognostic model in ref. 36 and the linear models in ref. 37 and the deep features extracted from the foundation model for both the NSCLC-Radiomics and NSCLC-Radiogenomics collections by the foundation model in ref. 37).

## Use Case I: deep learning for lung cancer prognostication

In the original study, Hosny et al. trained a CNN for predicting outcomes by analyzing tumor lesions on CT images and shared the model with the original investigation, together with the AI-derived prognostic scores for the patients analyzed in the study. The original publication used a subset of the NSCLC-Radiomics dataset[38,45] ($N = 211$ for the AUC analysis and $N = 307$ for the Kaplan–Meier analysis), as well as older and incomplete clinical and survival data. In our study, for both use cases, we use an updated dataset ($N = 422$) of the same cohort with clinical and follow-up data updated at the end of 2020 (NSCLC-Radiomics Version 4). Compared to the version used by Hosny et al., the dataset was substantially updated to address issues identified in earlier versions, such as corrections for the segmentation mask misalignments and omissions. In addition, the

clinical data was updated to reflect follow-up durations and survival information.

We provide an overview of the re-implementation in Fig. 3. Following the textual description provided by the authors, we implemented a preprocessing pipeline for converting the imaging data retrieved from IDC to a format suited for the deep-learning pipeline using open-source tools. To promote forward compatibility, we also converted the original model weights in the open neural network exchange (ONNX)[62]. We make the code for the model conversion and additional details available as part of our project repository, together with a notebook that incorporates a comprehensive description of every step of the replication process to help users navigate the complexity of the pipeline. Finally, we share a notebook with the code used to generate the results we present in this publication[63].

The comparison between the original pipeline and the re-implementation was carried out using several metrics, statistical, and survival analysis tools—such as the area under the receiver operating characteristic (ROC) curve (AUC), the two-sided Mann–Whitney $U$ test, the DeLong test for paired AUC curves, and Kaplan–Meier (KM) and Cox Proportional Hazard (PH) modeling—with the study concluding that deep-learning techniques could provide valuable insights into lung cancer prognosis and potentially assist in personalized treatment decisions.

## Use Case II: foundation models for quantitative biomarker discovery

In this study, Pai et al. investigated whether foundation models, pre-trained using self-supervised learning, can improve the development of deep-learning-based imaging biomarkers. The foundation model from the paper was validated on the NSCLC-Radiomics dataset and the NSCLC-Radiogenomics dataset[46,47]. Details for the former are available in the previous section, while the NSCLC-Radiogenomics includes CT images for 211 NSCLC stage I–IV patients acquired at the Stanford University School of Medicine and the Palo Alto Veterans Affairs Healthcare System. The dataset also features segmentation maps of the tumors, manually drawn from the CT scans, and semantic annotations of the tumors as observed on the medical images using a controlled vocabulary. In addition, the imaging information is matched with clinical and survival data. As in ref. 37, we included all the patients with annotated gross tumor volumes and clinical data ($N = 133$) in our study. Pai et al. concluded that modeling using deep features from the foundation model, despite being substantially more resource-efficient than supervised learning, was the most robust across tasks, offering stable performance even when the size of the fine-tuning datasets was considerably reduced.

The model validation was carried out using several metrics and survival analysis tools, such as the area under the receiver operating characteristic (ROC) curve (AUC), Kaplan–Meier (KM) analysis, and Cox Proportional Hazard (PH) modeling.

## Data availability

All of the imaging data used in the study is publicly available through the Imaging Data Commons platform. The aforementioned notebooks contain the query or manifest used to generate the exact subset of data used for analysis purposes. All of the clinical data is publicly available through the Imaging Data Commons platform and the TCIA[64]. The notebooks at the project repository contain the link to such clinical data and the code to generate the exact subset of data used for analysis purposes. All of the artifacts (e.g., the risk scores and the deep features) generated from the two pipelines are available in the project repository and as part of the Zenodo release.

## Code availability

The codebase and the relative documentation can be accessed at the project repository https://github.com/ImagingDataCommons/idc-

radiomics-reproducibility/releases/tag/v2.1.0 and referred to through Zenodo (https://zenodo.org/records/10123555). The cloud implementation of the pipeline can be accessed and run through the notebooks found at https://github.com/ImagingDataCommons/idc-radiomics-reproducibility/tree/main/notebooks. For Hosny et al.'s use case: The minimal working example of the cloud pipeline can be run at https://colab.research.google.com/github/ImagingDataCommons/idc-radiomics-reproducibility/blob/main/notebooks/hosny_processing_example.ipynb. The complete model validation cloud pipeline can be run at https://colab.research.google.com/github/ImagingDataCommons/idc-radiomics-reproducibility/blob/main/notebooks/hosny_complete_inference.ipynb. The results analysis, generating all of the plots used in the manuscript figures, can be run at https://colab.research.google.com/github/ImagingDataCommons/idc-radiomics-reproducibility/blob/main/notebooks/hosny_results_comparison.ipynb. For Pai et al.'s use case: The minimal working example of the cloud pipeline can be run at https://colab.research.google.com/github/ImagingDataCommons/idc-radiomics-reproducibility/blob/main/notebooks/pai_processing_example.ipynb. The complete model validation cloud pipeline, including the results analysis (generating all of the plots used in the manuscript figures), can be run at https://colab.research.google.com/github/ImagingDataCommons/idc-radiomics-reproducibility/blob/main/notebooks/pai_complete_inference.ipynb.

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

## Acknowledgements
This project has been funded in whole or in part with Federal funds from the NCI, NIH, under task order no. HHSN26110071 under contract no. HHSN261201500003l. The authors also acknowledge financial support from the European Union, European Research Council (H.A. grant no. 866504).

## Author contributions
Study conceptualization: D.B., H.A. and A.F. Code implementation and result analyses: D.B. and S.P. Writing of the manuscript: D.B., L.N., H.A., A.F. and D.K. Critical revision of the manuscript: D.B., L.N., H.A., A.F., D.K., V.T., A.H., R.M., K.F., R.K. and S.P. Study supervision: H.A. and A.F.

## Competing interests
The authors declare no competing interests.
