## [Peer Review File · Nature Communications]

End-to-End Reproducible AI Pipelines in Radiology Using the CloudREVIEWER COMMENTS

Reviewer #1 (Remarks to the Author):

The work presented by Dennis Bontempi and colleagues proposes an idea for how 3rd party cloud-computing infrastructure could be used to render AI-based (or AI-containing) Medical Imaging Pipelines reproducible. The idea is not new (see for example [1]) and the advantages of making oneself reliant on a third-party cloud computing provider might, in the opinion of this reviewer, much outweigh the advantages.

****Key results****

As a demonstration of the idea, the authors re-implemented an analysis workflow by Hosny et al., which is in fact a wonderful example of an openly available method that can operate on openly available data.

****Validity and Significance****

The reproduced analysis pipeline does what it should and is based on data available in the cloud (via NIH's Imaging Data Commons) and computed via the Google Colaboratory, a cloud-based notebook service offered by Google.

As I understand, the authors propose that medical imaging pipelines should more frequently be implemented, maintained, and publicly provided in the same technical way. This would ensure a more homogeneous landscape for computational methods and hence improve reproducibility (e.g. by removing issues that are a consequence of everyone using their favorite or most familiar tools).

While the presented pipeline does what it should, I see multiple issues with the presented work that the manuscript fails to answer.

- Who are the people that should follow the proposals in the manuscript and how are the authors planning to convince a critical mass of scientists to adopt their proposals? Without reaching critical mass, the proposed ideas will fail to deliver the sold benefits (since it would just be one of the many different and heterogeneous ways of openly providing reproducible pipelines).
- After looking at Hosny et al. (the pipeline reimplemented by the authors), I wondered: how much time would it really save me to familiarize myself with the reimplementation instead of re-running the original pipeline by Hosny et al. Also, they provide links to the data and to a code repository. Can the advantage be quantified? How big is the advantage for other medical pipelines?
- If, instead of proposing to use the Google Colaboratory, the authors would propose a common compute environment and set of (suggested) dependencies, wouldn't the set of advantages be essentially the same? Still, not depending on the environment and dependency constraints set up by the third party (Google), everyone who develops for this common environment enables a similarly convenient way to reproduce their work. Of course, someone would have to keep this joint environment up-to-date, but it stands to reason that Google's decisions (which are drawn from considerations that might be orthogonal to the ones relevant for the medical community) are less optimal than the ones the community itself would make.

****Data and methodology****

Presented data and computational pipeline have been made available prior to the presented work. The novelty of this work is limited to proposing a straightforward way to use openly and freely available third-party cloud computing infrastructure instead of running the openly available code on any suitable computing hardware (after, and this indeed requires some additional effort, installing required dependencies).

****Analytical approach****

Does not apply.

****Clarity and context****

The manuscript is well written and also outlines many of the problems one would expect by starting to rely on freely available cloud computing hard- and software.

****Related work that should be mentioned (more relevant work might exist)****

There are efforts that use freely available cloud-computing infrastructure to (re-)implement AI-containing computational modules. See [1].

There are efforts to make AI modules reproducible. See [2,3].

****Summary****

In summary, I am left to wonder if the advantages of committing to using a third-party hard- and software stack is the right way to go for a field. Yes, committing to a common environment removes the overheads of heterogeneity, but it can also be constraining and comes with the danger of making oneself dependent on decisions the third party draws (problems as the ones eloquently enumerated in the manuscript).

The manuscript contains one rather obvious idea, demonstrated on a single example pipeline. Other published work of similar nature exists but adds additional value to readers/users by providing reimplementations of many very useful computational AI modules [1].

At the same time, other approaches [2,3] to making AI modules themselves reproducible (independent from the surrounding pipeline). At the very least I would have hoped to see the AI parts in the presented work to pick up such alternative efforts.

Personally, I don't see sufficient novelty to justify publication. Even worse, I remain with the belief that the benefits are much out-weight by the dangers of making an entire (sub-)community reliant on the uncontrollable decisions of a third party (Google).

*****References:*****

[1] <https://www.nature.com/articles/s41467-021-22518-0>

[2] <https://monai.io/model-zoo.html>

[3] [<https://bioimage.io/>](<https://bioimage.io/#/>) - <https://www.biorxiv.org/content/10.1101/2022.06.07.495102v1>

Reviewer #2 (Remarks to the Author):

In this article, the authors focus on a common issue within medical research wherein AI methodologies suffer from a lack of transparency and reproducibility. This problem comes from the many design choices and hyperparameters involved, which are not consistently shared across studies. The authors present the hypothesis that utilizing cloud-hosted data may alleviate this issue, as models would not reside on local computers but within a shared cloud environment, thereby facilitating easier reproduction.

While this proposition is not novel and has been previously suggested, its widespread adoption remains limited, perhaps due to a lack of systematic evidence. The authors recognize this gap in the existing literature and strive to investigate it, which is commendable.

The paper is a blend of a conceptual proposal in a perspective style and an original research paper. It would definitely benefit from leaning more towards the research paper format by presenting clear and measurable hypotheses, delineating explicit experimental designs to test these hypotheses, and presenting the results in a clear manner. The current structure lacks conventional

headings, reminiscent more of a brief communication paper. To enhance readability, it would be good to follow the classical division of sections into introduction, methods, results, and discussion. Furthermore, it would be advisable for the authors to down-tone their language, as many portions of the manuscript tend towards the speculative and promotional rather than maintaining a scientific tone.

The authors do not sufficiently discuss the limitations of their approach. For example, in my experience leading a large research group focusing on medical imaging analysis, we have found that most PhD students, even those proficient in computer science, prefer downloading data to a local machine. This preference stems from the flexibility and freedom offered by local computing, allowing the immediate application of the latest tools and methodologies. The administrative and logistical burdens associated with cloud-based computing, such as writing proposals and waiting for approvals, inhibit rapid experimentation and innovation, which are essential in our fast-paced research environment.

The authors have also neglected to consider two additional significant limitations. Firstly, the immense data requirements of certain medical image analysis problems, as illustrated by the UK Biobank's dealing with 50,000 brain MRI images, often exceed the practical capabilities of conventional cloud-based resources. Secondly, and perhaps most crucially, they have not fully recognized the institutional reluctance to share medical data via cloud platforms. This resistance often necessitates data to remain on-premise, making an approach such as federated learning more viable. This should be discussed.

Finally, the figures included within the article require revision to be more scientific. As they stand, they resemble components of a business pitch rather than scientifically substantiated data representations. For the paper to resonate with the scientific community, these figures should be crafted with greater substance and precision and contain more actual data as opposed to concepts.

Reviewer #3 (Remarks to the Author):

This study proposes a practical approach and strategy for a relevant and important challenge existing nowadays in the field of AI solutions for medical imaging. It clearly shows that the use of cloud infrastructures and services are benefitting to increased transparency and reproducibility of studies on AI models for radiology or medical imaging. Another important message is that this type of resources are available and accessible for small teams and low-budget research entities or clinicians-engineers units, which provide a solution for the democratization and wide adoption of these AI solutions. Finally, the ease of access and use of these tools is also a major result contributing to the main aim of the authors who rightly demonstrate that the skill gap could be easily overcome.

The results from the data analysis in this study appear valid and robust. The various techniques and statistical analysis have demonstrated the replicability of the results from the original study and use-case. The demonstration is sound and described in detail.

Medical imaging is one of the most advanced medical domains in terms of development of AI for healthcare solutions. The market is dynamic, showing an increasing number of new players (large industries and startups) proposing solutions for clinical practices. The economic stakes are, in this context, very high. Nevertheless, the adoption in clinical practices is often mentioned as a barrier and the question of trust from health professionals (radiologists or other physicians) and the large public (patients) is always key for the deployment of such solutions. Several policymakers have taken initiatives to define legal and ethical frameworks around AI technologies, especially in the medical domain. The capacity to deploy methodologies and practical guidance to increase the transparency and the reproducibility contributes to these efforts, allowing independent researchers and/or regulatory bodies to operate audits and monitoring on AI solutions proposed to the market. This study is thus highly relevant to the field of AI in healthcare globally, but as well to medicine at large given the expected major role of these technologies on medical practices in the upcoming years.

The methodology is sound. The presented data in the article as well as in the supplementary information are well presented and detailed information and content is provided (in annex) around the AI pipeline process. The notebooks information is specifically helping to clarify and present the benefit of the use of such cloud-based tools and illustrate perfectly the gains in terms of support and reproducibility for researchers willing to execute and verify the approach of the original study. The analytical approach is pretty classic and basic but serves correctly the purpose of the study and demonstrate soundly the hypothesis formulated by the authors.

Here are some suggestions to improve the content of this article, both in terms of description of the proposed approach and in terms of discussion of the results:

- It could be interesting for the reader to have access to a list of potential alternative tools to Google Collaboratory, highlighting the most important key features of such platforms that are required to develop the proposed approach for AI pipelines documentation and management.
- Mention and refer to the applicability of the proposed cloud-based approach when using synthetic data or differential privacy approaches applied on existing private datasets, which are increasing a frequent data context guarantying a better level of privacy for patients.

The authors present their approach as a solution for the lack of transparency of AI scientists or engineers when developing studies on new models; this behaviour hindering the reproducibility of the the experiments. Nevertheless, beyond the use of appropriated tools, such as cloud-based solutions as described in the article, other reasons could have been mentioned for this lack of transparency, such as:

- Question of Intellectual Property Rights (IPR) on development of new AI models (especially in the case of models aimed to exploited in a commercial manner
- Ai for healthcare is a very competitive market, and most of the AI models developers for medical imaging face a challenge of speed of development and deployment. They might be reluctant to transparency in order to keep their competitive advantage. Most of the AI researchers/engineers teams are requested to "go fast" and so don't take time to make available all information for transparency and reproducibility
- Also, as a large part of current AI research in medical imaging are driven and funded by industry players, the researchers might not be authorized to publish everything.
- Many startups are currently developing AI models for medical imaging and their R&D teams might not be compliant with the mentioned Ai development guidelines because of their lack of knowledge or experience on these guidelines.
- The clients on this market (radiologists) are not very demanding in terms of scientific proofs and transparency/reproducibility so the developers are not prioritizing the tasks enabling full transparency and reproducibility.

The references of this article about previous literature are appropriated and reflecting the state of the knowledge around the targeted domain.

Point-by-Point Response to the Reviewers' Comments

We thank the reviewers for their comprehensive feedback. We believe their comments contributed to substantial improvements in our manuscript, including new experiments, deeper analyses, and refined text, enriching the quality of our work.

We would like to address some general comments here:

- 1. New experiments:** As part of this revision, we included a second use case to further validate the proposed workflow. We share the validation of a novel self-supervised model for biomarker discovery in oncology (from Pai et al. - Foundation Models for Quantitative Biomarker Discovery in Cancer Imaging [1]) using cloud-based computational resources and cloud-hosted data. Pai et al. developed a foundation model for imaging biomarker discovery on an extensive dataset of radiographic lesions. The model offers a robust and reliable framework for discovering cancer imaging biomarkers, even in small datasets, surpassing current deep learning techniques in various tasks while fitting conveniently into existing radiomic research methods. We believe the replication of this paper using our framework can be of great interest, as the resources we share can be used not only to verify Pai et al.'s findings but also to understand how the pipeline can be extended upon and used for other studies. We hope this significant addition can help the editor and the reviewers see the value of the proposed workflow for disseminating novel, transparent, and reproducible research and its potential benefits to the radiology imaging community and the broader cancer imaging research community.
- 2. Manuscript structure and target:** We restructured the manuscript in a canonical format. The introductory section was updated based on the addition of a second use case, the main text was formatted with a sub-heading and a clear distinction between methods and results, and the tone of the paper has been adjusted as per the editor's and reviewers' remarks. Furthermore, we acknowledge we previously failed to highlight the novelty of the workflow proposed in the manuscript by mistakenly addressing our contribution as helpful for the broad medical imaging community instead of the more narrow radiology community. Being the use cases included in our work, together with the tools and best practices we share, focused exclusively on radiology, in the revised manuscript, we changed the term "medical imaging" to "radiology" wherever "medical imaging" could read misleadingly. To make the contribution of our work clearer, we also changed the title of the manuscript to "End-to-End Reproducible AI Pipelines in Radiology Using the Cloud." To the best of our knowledge, we are the first to propose such a cloud-based pipeline that could pragmatically help such a community with reproducibility and transparency. As pointed out by one reviewer, similar efforts were previously developed - but never in this field (e.g., the publications referenced targeted digital microscopy) and, most importantly, never connecting public data sources storing hundreds of thousands of image series to cloud-based computational resources - making the whole workflow entirely transparent, reproducible, and residing completely on the cloud. We therefore believe those other efforts are completely orthogonal from ours.

We updated the revised manuscript to incorporate these important new insights and data. A detailed point-by-point response to the reviewers' comments can be found below, where we responded in blue.

[1] Pai et al. - Foundation Models for Quantitative Biomarker Discovery in Cancer Imaging. medRxiv (2023). <https://doi.org/10.1101/2023.09.04.23294952>

Reviewer #1

Remarks to the Author:

The work presented by Dennis Bontempi and colleagues proposes an idea for how 3rd party cloud-computing infrastructure could be used to render AI-based (or AI-containing) Medical Imaging Pipelines reproducible. The idea is not new (see for example [1]) and the advantages of making oneself reliant on a third-party cloud computing provider might, in the opinion of this reviewer, much outweigh the advantages.

We thank the reviewer for raising this point. However, we respectfully disagree and would like to argue why we believe the drawbacks of relying on a third-party cloud computing provider do not outweigh the advantages.

Although we agree that relying on a third-party cloud computing provider might, in some aspects, be limiting, we believe that:

- It significantly improves how AI-based computational pipelines for radiology are shared. As we discuss in our work, more often than not, at least a component is missing or not working as intended, and therefore, the pipeline cannot be reproduced. We believe written guidelines are not enough to address all of the intricacies that such pipelines intrinsically bear.
- It is, to the best of our knowledge, the only way to democratize access to these resources by providing a free, fully-functioning version of a computational pipeline (something that, given the costs of the infrastructure and all of the software components, needs a cloud-service provider to be implemented efficiently and free-of-charge).
- There are no solutions to addressing reproducibility that would not have any limitations.

Furthermore, we would like to stress the fact that it is not in our interest to force a user to pick a specific cloud provider over another. In von Chamier et al. [1], the authors show that the same pipelines can be implemented and executed using different cloud providers. Although we believe this makes such resources harder to maintain, it proves that the proposed blueprint does not rely on a specific third-party cloud computing provider. Moreover, as we explain in depth below (answering another comment), we show the resources we share can be used locally, independently from the third-party cloud provider of choice.

[1] von Chamier et al. - Democratising deep learning for microscopy with ZeroCostDL4Mic. Nat Commun 12, 2276 (2021). <https://doi.org/10.1038/s41467-021-22518-0>

****Key results****

As a demonstration of the idea, the authors re-implemented an analysis workflow by Hosny et al., which is in fact a wonderful example of an openly available method that can operate on openly available data.

We thank the reviewer for these positive remarks.

Expanding on this, we added to the revised manuscript and the analysis a novel use case (part of a paper under review) that ought to demonstrate how the proposed workflow can help with the dissemination of new results and strengthen transparency and reproducibility for new findings - as opposed to previously published literature.

****Validity and Significance****

The reproduced analysis pipeline does what it should and is based on data available in the cloud (via NIH's Imaging Data Commons) and computed via the Google Colaboratory, a cloud-based notebook service offered by Google.

As I understand, the authors propose that medical imaging pipelines should more frequently be implemented, maintained, and publicly provided in the same technical way. This would ensure a more homogeneous landscape for computational methods and hence improve reproducibility (e.g. by removing issues that are a consequence of everyone using their favorite or most familiar tools).

We thank the reviewer for their remark. We indeed believe the proposed workflow can ensure a more homogeneous landscape for computational methods, and we are also convinced this effort can have a significant impact on the training and education of the new generation of computational scientists.

As mentioned before, we believe the proposed framework can help democratize access to state-of-the-art AI-based computational methods for radiology.

While the presented pipeline does what it should, I see multiple issues with the presented work that the manuscript fails to answer.

- Who are the people that should follow the proposals in the manuscript and how are the authors planning to convince a critical mass of scientists to adopt their proposals? Without reaching critical mass, the proposed ideas will fail to deliver the sold benefits (since it would just be one of the many different and heterogeneous ways of openly providing reproducible pipelines).

We thank the reviewer for the insightful comment. We agree that not achieving a critical mass of adopters might hinder some of the benefits of the proposed framework. However, we also believe in the intrinsic value of high-quality use cases, even if they are limited in number. A smaller set of well-constructed, comprehensive use cases can serve as foundational pillars for education and training in the medical image analysis field - as proven already in other areas, such as Genomics research [1], since:

- These use cases can set a standard of quality and rigor for others to aspire to or compare against, raising the overall quality of research in the field.
- For new practitioners in the field of AI in medical imaging, having access to a few meticulously crafted use cases can offer an in-depth understanding of the subject matter, methodologies, and, most importantly, best practices. We believe this could be more beneficial than having many use cases of variable quality.
- High-quality use cases have the potential to attract a core community of dedicated researchers who are passionate about transparency and reproducibility. This core group can then drive further adoption and expand the framework to suit specific needs.

Although we agree the proposed framework may not be adopted by many in a short period of time, we believe it has more potential to help transparency and reproducibility in the field than many text-based recommendations or guidelines that are found in the literature.

[1] Ben Guebila et al. - An online notebook resource for reproducible inference, analysis and publication of gene regulatory networks. Nat Methods 19, 511–513 (2022). <https://doi.org/10.1038/s41592-022-01479-2>

- After looking at Hosny et al. (the pipeline reimplemented by the authors), I wondered: how much time would it really save me to familiarize myself with the reimplementation instead of re-running the original pipeline by Hosny et al. Also, they provide links to the data and to a code repository. Can the advantage be quantified? How big is the advantage for other medical pipelines?

We thank the reviewer for raising this point. This is a crucial aspect of why the proposed framework was developed in the first place, and we may have failed to stress enough in the original submission.

While it's challenging to precisely quantify the time a user can save using the resources shared by our work (and implemented following the proposed framework), we believe that such a standardized process substantially reduces the time the user would spend deciphering, reinventing, or debugging common tasks.

For instance, should a user (which we assume familiar with the field, for the sake of simplicity) want to set up everything necessary to run Hosny et al. pipeline on local hardware to understand whether the model can be useful for their application, such a user would be required to:

- Set up a use-case-dependent environment (e.g., python libraries and system dependencies)
- Assuming the user already has medical imaging and clinical data that suit the use case and can provide a reliable benchmark for the model, write the code to prepare the data for processing. This usually requires the user to reimplement use-case-specific data preparation steps, often omitted in the codebase (as in our case), which tends to be time-consuming and has the potential to introduce inconsistencies in the results (since the plethora of tools available to implement such steps often differs in some key aspects)
- Familiarize themselves with the codebase, the resources, and the documentation shared by the authors (if any). Depending on the complexity of the pipeline in question, this might require a significant amount of time.
- Assuming everything works as intended after the set-up and the data processing, re-implement a result evaluation pipeline to reproduce or extend the findings of the original publication. This often relies, again, on the choice of a preferred tool among many that might require specific expertise to be run as intended (e.g., it is often the case for a deep learning practitioner tasked with running a pipeline on in-house data not to be entirely familiar with the analyses published in a medical study, since many studies lie at the intersection between different fields)

To expand on these points, although we are conscious this is an anecdotal experience, the re-implementation of the pipeline was not straightforward despite having a connection to the authors of the original publication (Hosny et al.). Hosny et al.'s codebase lacked instructions in everything concerning the data conversion and pre-processing. We provide an example in the "data exploration" section of the complete inference notebook in our project repository under "notebooks." Furthermore, as reported in the documentation under `src/model/` in our project repository, several steps were needed to make the model function as intended using libraries that are not deprecated. Finally, none of the statistical analysis code was shared by the authors.

Overall, we feel these hurdles (i.e., not having something that works off-the-shelf, but rather, having to invest a significant amount of time to make pipelines work) more often than not can discourage researchers from expanding on existing studies. Rather than researchers spending time 'reinventing the wheel' with each new project, we believe they would greatly benefit from resources developed following our cloud-based workflow. This would not only allow them to save hours of coding, set-up, and potential debugging - but it would also enable such users to focus their efforts on the application and extension of the AI pipelines in question.

Following the reviewer's remark, we added a section in the Supplementary Information ("Practical Differences Compared to a Local Setup," under "Supplementary Methods") to address similar concerns readers of the paper will likely have.

- If, instead of proposing to use the Google Colaboratory, the authors would propose a common compute environment and set of (suggested) dependencies, wouldn't the set of advantages be essentially the same? Still, not depending on the environment and dependency constraints set up by the third party (Google), everyone who develops for this common environment enables a similarly convenient way to reproduce their work. Of course, someone would have to keep this joint environment up-to-date, but it stands to reason that Google's decisions (which are drawn from considerations that might be orthogonal to the ones relevant for the medical community) are less optimal than the ones the community itself would make.

We thank the reviewer for raising this critical point. We believe maintaining such a computing environment would bear maintenance efforts that are significantly more costly, in terms of time, than developing resources following the proposed workflow. Therefore, a possible solution to address the reviewer's concerns would be to make use of a predefined computational environment as an alternative to the proposed workflow, such that the resources we share could be run locally as well as in the cloud. Inspired by this, we decided to set up a local computational instance using the publicly available Docker images upon which Google Colab is based and re-run all of the experiments we shared in the repository. It is rather critical to note that the ability to use Docker depends on the availability of the infrastructure and resources provided by Docker Inc. - a for-profit company funded by venture capital. In turn, Docker infrastructure relies on commercial cloud providers to deliver its content to the users. As demonstrated, we succeeded in eliminating the reliance on Google Colab, but there is no solution that would achieve a complete third-party-free alternative to run the resources we share as part of our project repository.

Following the reviewer's remark, we added a section in the Supplementary Information ("Using Cloud-based Resources with Local Runtimes," under "Supplementary Methods") to address similar concerns readers of the paper will likely have. Furthermore, we added a section to the repository's documentation guiding the user through how to run the notebooks using the aforementioned computational environment (under "running the pipeline locally").

We strongly believe using the cloud is the most immediate and easy solution, but we hope this alternative can satisfy the reviewer's concerns. We agree relying on a third party has its disadvantages (as already pointed out above). However - even if it could be helpful in a different manner - using such a common computing environment and a set of dependencies to run such experiments does not allow the user to run and inspect a pipeline for free (which is, on the other hand, enabled by the proposed workflow) and can still require significant efforts to set up (e.g., Docker) and/or expensive hardware to be bought beforehand (such as GPU-equipped machines). We believe this substantially hinders the assessment of transparency and reproducibility of said pipeline by reviewers and practitioners, the sharing of best practices for specific computational steps, and so on.

****Data and methodology****

Presented data and computational pipeline have been made available prior to the presented work. The novelty of this work is limited to proposing a straightforward way to use openly and freely available third-party cloud computing infrastructure instead of running the openly available code on any suitable computing hardware (after, and this indeed requires some additional effort, installing required dependencies).

We respectfully disagree with the reviewer in regard to this point.

As extensively mentioned above, there are several steps that are often overlooked by researchers willing to share their medical image analysis pipeline that very often hinder the application and external validation of such pipelines (as discussed in one of our answers above and in “Practical Differences Compared to a Local Setup” in the Supplementary Information). Furthermore, we believe another advantage of the proposed workflow lies in the use of the cloud-based data part of the Imaging Data Commons (IDC), which are harmonized, versioned, and easy to access.

Finally, we believe a computational environment that is as user-friendly as Colab makes the integration of educational material (from the simple explanation of what operations are being run by a specific code snippet to the in-browser data and results visualization) seamless and easy to digest.

****Analytical approach****

Does not apply.

****Clarity and context****

The manuscript is well written and also outlines many of the problems one would expect by starting to rely on freely available cloud computing hard- and software.

We thank the reviewer for these positive remarks.

****Related work that should be mentioned (more relevant work might exist)****

There are efforts that use freely available cloud-computing infrastructure to (re-)implement AI-containing computational modules. See [1].

There are efforts to make AI modules reproducible. See [2,3].

We thank the reviewer for their remarks. We believe resources such as Model Zoos [2][3] are very helpful for collecting pipelines, often trained following a specific paradigm or using a specific library (as for [2]), but fail in providing the user a blueprint to make research transparent and reproducible. For instance, should a researcher take on the task of running a model from [2] or [3], they would at least need to look into what data needs to be procured, how that data needs to be pre-processed, and how could the model results be evaluated. In our experience, these steps can be opaque and often discourage the community from applying these models further. Most notably, model zoos are built to host models belonging to a certain community (e.g., MONAI) and not to run and inspect pipelines. The proposed workflow, which connects cloud-hosted data to computational resources and educational material, is devised to provide the user with something that is ready to be run and shared without leaving the user the need to connect all of the components.

As an additional example and to further clarify the difference between the proposed workflow and model zoos and guidelines, we added another use case. Following the workflow, we share the validation of a novel self-supervised model for biomarker discovery in oncology [5], packaged and ready to be run in a completely transparent and reproducible manner on the cloud. Given the complexity of the pipeline, we believe that should the developers share this model in a model Zoo, it would take the users substantially longer to familiarize themselves with the pipeline and expand this work. Furthermore, we believe sharing the model as part of a collection would not be sufficient to enable the complete reproducibility of the analysis. On the other hand, by implementing the validation pipeline following the proposed workflow, the users can run the end-to-end analysis in a matter of minutes without the need to code any data preparation and model evaluation, saving the effort of connecting the computational model to the data component.

Finally, although we agree with the reviewer, there have been efforts in sharing computational tools and modules to help with reproducibility [1], to the best of our knowledge, the proposed workflow is the first to connect the data to the computational and analysis pipelines, and doing so in the field of radiology.

In light of the reviewer's comments, we expanded the discussion section to include further elaboration on the scope of this work compared to the aforementioned literature[1][2][3].

[1] von Chamier et al. - Democratising deep learning for microscopy with ZeroCostDL4Mic. Nat Commun 12, 2276 (2021). <https://doi.org/10.1038/s41467-021-22518-0>

[2] MONAI Model Zoo. <https://monai.io/model-zoo.html>

[3] Ouyang et al. - BiImage Model Zoo: A Community-Driven Resource for Accessible Deep Learning in BiImage Analysis. BioRxiv (2022). <https://doi.org/10.1101/2022.06.07.495102>

****Summary****

In summary, I am left to wonder if the advantages of committing to using a third-party hard- and software stack is the right way to go for a field. Yes, committing to a common environment removes the overheads of heterogeneity, but it can also be constraining and comes with the danger of making oneself dependent on decisions the third party draws (problems as the ones eloquently enumerated in the manuscript).

The manuscript contains one rather obvious idea, demonstrated on a single example pipeline. Other published work of similar nature exists but adds additional value to readers/users by providing reimplementations of many very useful computational AI modules [1].

At the same time, other approaches [2,3] to making AI modules themselves reproducible (independent from the surrounding pipeline). At the very least I would have hoped to see the AI parts in the presented work to pick up such alternative efforts.

Personally, I don't see sufficient novelty to justify publication. Even worse, I remain with the belief that the benefits are much out-weight by the dangers of making an entire (sub-)community reliant on the uncontrollable decisions of a third party (Google).

We thank the reviewer for their remarks. We recognize the effort presented in our paper benefits primarily the radiology field and apologize for the confusion this might have caused. We hope the title

change and the various edits in the manuscript help highlight the difference between our workflow and [1] (which has a strong focus on microscopy and training).

As we mentioned in other parts of this document, we are convinced the workflow we present holds different values compared to [2] and [3]. We believe simply including a model as part of a collection or a model zoo does not ensure reproducibility and, more often than not, does not help with transparency. A model can be part of a Zoo and yield unexpected results, be difficult to set up, and, most importantly, require local hardware and a specific environment setup and dependencies to be run as intended (e.g., to verify the results of a study). Furthermore, should the user go to the length of setting up everything necessary to run a model from a zoo, this would still not make a pipeline easy to share in its entirety, as the data component and the possibility to run the end-to-end analysis is still missing.

]As for the last point, we clarified that the resources we provide could be run locally as well and provided a publicly available computational environment (and instructions) to do so, as the reviewer suggested.

[1] von Chamier et Aa. - Democratising deep learning for microscopy with ZeroCostDL4Mic. Nat Commun 12, 2276 (2021). <https://doi.org/10.1038/s41467-021-22518-0>

[2] MONAI Model Zoo. <https://monai.io/model-zoo.html>

[3] Ouyang et al. - BioImage Model Zoo: A Community-Driven Resource for Accessible Deep Learning in BioImage Analysis. BioRxiv (2022). <https://doi.org/10.1101/2022.06.07.495102>

Reviewer #2

Remarks to the Author:

In this article, the authors focus on a common issue within medical research wherein AI methodologies suffer from a lack of transparency and reproducibility. This problem comes from the many design choices and hyperparameters involved, which are not consistently shared across studies. The authors present the hypothesis that utilizing cloud-hosted data may alleviate this issue, as models would not reside on local computers but within a shared cloud environment, thereby facilitating easier reproduction.

While this proposition is not novel and has been previously suggested, its widespread adoption remains limited, perhaps due to a lack of systematic evidence. The authors recognize this gap in the existing literature and strive to investigate it, which is commendable.

We thank the reviewer for these positive remarks.

The paper is a blend of a conceptual proposal in a perspective style and an original research paper. It would definitely benefit from leaning more towards the research paper format by presenting clear and measurable hypotheses, delineating explicit experimental designs to test these hypotheses, and presenting the results in a clear manner. The current structure lacks conventional headings, reminiscent more of a brief communication paper. To enhance readability, it would be good to follow the classical division of sections into introduction, methods, results, and discussion. Furthermore, it would be advisable for the authors to downtone their language, as many portions of the manuscript tend towards the speculative and promotional rather than maintaining a scientific tone.

We thank the reviewer for these suggestions, and we agree with the assessment. In the revised manuscript, we have structured/positioned the manuscript as a research paper that includes a defined hypothesis, as well as structured with an introduction, results, discussion, and methods. Furthermore, the tone of the paper has been adjusted as per the reviewer's remark.

The authors do not sufficiently discuss the limitations of their approach. For example, in my experience leading a large research group focusing on medical imaging analysis, we have found that most PhD students, even those proficient in computer science, prefer downloading data to a local machine. This preference stems from the flexibility and freedom offered by local computing, allowing the immediate application of the latest tools and methodologies. The administrative and logistical burdens associated with cloud-based computing, such as writing proposals and waiting for approvals, inhibit rapid experimentation and innovation, which are essential in our fast-paced research environment.

We thank the reviewer for their remarks. We agree with their assessment that more limitations needed to be discussed (which has been included in the revised discussion), but this does not imply that cloud-based pipelines are not helpful for the field. While we agree rapid experimentation and innovation (e.g., training a model from scratch) can be carried out locally, we believe sharing cloud-based pipelines following the proposed workflow can not only bolster the reproducibility of published literature but also serve as a foundational pillar for education and training in the medical image analysis field - as proven already in other areas, such as Genomics research [1]. In other words, we believe that accompanying research carried out locally with cloud-based resources connecting imaging and clinical data, computational tools, and results analysis tools can be beneficial for the field.

For instance, these resources can set a standard of quality and rigor for others to aspire to or compare against, raising the overall quality of research. Furthermore, for new practitioners in the field of AI in medical imaging, having access to a few meticulously crafted use cases can offer an in-depth understanding of the subject matter, methodologies, and, most importantly, best practices.

Moreover, as part of this revision, we provide a way to experiment with the resources we shared locally. We set up a local computational instance using the publicly available Docker images upon which Google Colab is based and re-run all of the experiments we shared in the repository. We succeeded in doing so and have added a section to the repository's documentation guiding the user through how to run the notebooks using the aforementioned computational environment (under "running the pipeline locally"). Although we strongly believe using the cloud is the most immediate and easy solution (as it doesn't require setup of any kind nor readily available hardware), we hope this alternative can satisfy the reviewer's concerns.

Following the reviewer's remark, we added a subsection in the Supplementary Information ("Using Cloud-based Resources with Local Runtimes," under Supplementary Methods) to address similar concerns readers of the paper will likely have.

[1] Ben Guebila et al. - An online notebook resource for reproducible inference, analysis, and publication of gene regulatory networks. *Nat Methods* 19, 511–513 (2022). <https://doi.org/10.1038/s41592-022-01479-2>

The authors have also neglected to consider two additional significant limitations. Firstly, the immense data requirements of certain medical image analysis problems, as illustrated by the UK Biobank's dealing with 50,000 brain MRI images, often exceed the practical capabilities of conventional cloud-based resources. Secondly, and perhaps most crucially, they have not fully recognized the institutional reluctance to share medical data via cloud platforms. This resistance often necessitates data to remain on-premise, making an approach such as federated learning more viable. This should be discussed.

We thank the reviewer for their remark - however, we respectfully disagree with them as the high scalability of cloud resources is one of the advantages as compared to the conventional on-premises resources (e.g., it may be physically or practically impossible to maintain resources of a very large scale at a single geographical location).

Taking the UK Biobank example, the project already has a cloud-based platform dedicated entirely to computational research. This resource is accessible at <https://ukbiobank.dnanexus.com/> and allows researchers to run any sort of computation entirely on the cloud (with their documentation reporting examples of use cases developed entirely using the platform). Furthermore, the NCI Imaging Data Commons (IDC) hosts, to this day, nearly 500,000 image series and is powering several cloud-based research studies that would be challenging, from a computational point of view, to run on-premises resources.

We would also like to stress the proposed workflow is something we find worth considering to enable easier sharing of research (and not only to run studies entirely on the cloud). For instance, a resource developed following the proposed workflow could accompany bigger analyses run locally on data that, for size* or privacy reasons, cannot be shared. Furthermore, we do believe that the proposed workflow can significantly lower the barriers to entering the medical image analysis field while offering a pragmatic way to address the transparency and reproducibility problems the field suffers from. In our experience, presenting the user with a common computational workflow, a pipeline that follows

published guidelines, or even a thoroughly documented codebase does not lower said entry barriers enough for a method to be tested or validated by the community.

To exemplify further how this might work, we added another use case to the revised manuscript . Following the workflow, we share the validation of a novel self-supervised model for biomarker discovery in oncology [1], ready to be run in a completely transparent and reproducible manner on the cloud - where all of the development and the analyses have been run locally (e.g., due to the impossibility of sharing some data publicly), with only the validation of the model being implemented following the workflow.

Regarding federated learning, we agree with the reviewer that sharing medical data via cloud platforms is not always feasible. To reflect this, we expanded the Discussion section (which already contained mention of this remark).

[1] Pai et al. - Foundation Models for Quantitative Biomarker Discovery in Cancer Imaging. medRxiv (2023). <https://doi.org/10.1101/2023.09.04.23294952>

Finally, the figures included within the article require revision to be more sciency. As they stand, they resemble components of a business pitch rather than scientifically substantiated data representations. For the paper to resonate with the scientific community, these figures should be crafted with greater substance and precision and contain more actual data as opposed to concepts.

We agree with the reviewer and have updated all figures to address their concerns.

Reviewer #3

Remarks to the Author:

This study proposes a practical approach and strategy for a relevant and important challenge existing nowadays in the field of AI solutions for medical imaging. It clearly shows that the use of cloud infrastructures and services are benefitting to increased transparency and reproducibility of studies on AI models for radiology or medical imaging. Another important message is that this type of resources are available and accessible for small teams and low-budget research entities or clinicians-engineers units, which provide a solution for the democratization and wide adoption of these AI solutions. Finally, the ease of access and use of these tools is also a major result contributing to the main aim of the authors who rightly demonstrate that the skill gap could be easily overcome.

The results from the data analysis in this study appear valid and robust. The various techniques and statistical analysis have demonstrated the replicability of the results from the original study and use-case. The demonstration is sound and described in detail.

We thank the reviewer for these positive remarks.

Medical imaging is one of the most advanced medical domain in terms of development of AI for healthcare solutions. The market is dynamic, showing an increasing number of new players (large industries and startups) proposing solutions for clinical practices. The economic stakes are, in this context, very high. Nevertheless, the adoption in clinical practices is often mentioned as a barrier and the question of trust from health professionals (radiologists or other physicians) and the large public (patients) is always key for the deployment of such solutions. Several policymakers have taken initiatives to define legal and ethical frameworks around AI technologies, especially in the medical domain. The capacity to deploy methodologies and practical guidance to increase the transparency and the reproducibility contributes to these efforts, allowing independent researchers and/or regulatory bodies to operate audits and monitoring on AI solutions proposed to the market. This study is thus highly relevant to the field of AI in healthcare globally, but as well to medicine at large giving the expected major role of these technologies on medical practices in the upcoming years.

We thank the reviewer for these positive remarks.

The methodology is sound. The presented data in the article as well as in the supplementary information are well presented and detailed information and content is provided (in annex) around the AI pipeline process. The notebooks information is specifically helping to clarify and present the benefit of the use of such cloud-based tools and illustrate perfectly the gains in terms of support and reproducibility for researchers willing to execute and verify the approach of the original study. The analytical approach is pretty classic and basic but serves correctly the purpose of the study and demonstrate soundly the hypothesis formulated by the authors.

We thank the reviewer for these positive remarks.

Here are some suggestions to improve the content of this article, both in terms of description of the proposed approach and in terms of discussion of the results:

- It could be interesting for the reader to have access to a list of potential alternative tools to Google Collaboratory, highlighting the most important key features of such platforms that are required to develop the proposed approach for AI pipelines documentation and management.

- Mention and refer to the applicability of the proposed cloud-based approach when using synthetic data or differential privacy approaches applied on existing private datasets, which are increasing a frequent data context guarantying a better level of privacy for patients.

We thank the reviewer for this observation. We agree with the reviewer that several cloud platforms exists besides Google Collaboratory. These important points were added to the Discussion section of the revised manuscript, where we mention the main alternatives to Colab and expanded on other aspects of privacy-preserving pipelines.

The authors present their approach as a solution for the lack of transparency of AI scientists or engineers when developing studies on new models; this behaviour hindering the reproducibility of the the experiments. Nevertheless, beyond the use of appropriated tools, such as cloud-based solutions as described in the article, other reasons could have been mentioned for this lack of transparency, such as:

- Question of Intellectual Property Rights (IPR) on development of new AI models (especially in the case of models aimed to exploited in a commercial manner
- AI for healthcare is a very competitive market, and most of the AI models developers for medical imaging face a challenge of speed of development and deployment. They might be reluctant to transparency in order to keep their competitive advantage. Most of the AI researchers/engineers teams are requested to “go fast” and so don’t take time to make available all information for transparency and reproducibility
- Also, as a large part of current AI research in medical imaging are driven and funded by industry players, the researchers might not be authorized to publish everything.
- Many startups are currently developing AI models for medical imaging and their R&D teams might not be compliant with the mentioned AI development guidelines because of their lack of knowledge or experience on these guidelines.
- The clients on this market (radiologists) are not very demanding in terms of scientific proofs and transparency/reproducibility so the developers are not prioritizing the tasks enabling full transparency and reproducibility.

We thank the reviewer for their remarks. We agree with the reviewer the points raised can also limit transparency (and reproducibility) in the field of AI. We expanded the discussion section of the revised manuscript according to their suggestions, including a discussion on the impact of IPR, the competition in the AI healthcare market, industry-funded projects on cancer imaging research.

The references of this article about previous literature are appropriated and reflecting the state of the knowledge around the targeted domain

We thank the reviewer for their positive remarks.

REVIEWERS' COMMENTS

Reviewer #1 (Remarks to the Author):

The revised manuscript exhibits significant improvements in response to the feedback from all three reviewers. The focus on the timely and important topic of reproducibility in computational pipelines is commendable and currently a subject of global discussion.

However, the manuscript has narrowed its scope to primarily address radiologists. This change, seemingly in response to the critiques raised in the reviews, comes across more as a superficial remedy rather than a comprehensive solution addressing the fundamental issues highlighted.

A notable concern, from my perspective, is the feasibility of implementing the proposed solutions for accessible and reproducible analysis pipelines across different regions. For instance, while the United States shows openness to using cloud providers for storing and processing scientific data, the stringent GDPR regulations in Europe and other data storage restrictions might impede the global applicability of the presented work.

The authors' intentions are certainly praiseworthy, and the subject matter is both timely and significant. However, the solutions presented seem rather improvised. While this does not inherently undermine their value, a more comprehensive solution would likely involve intricate global policies and political considerations. Current efforts involving governmental and scientific bodies are actively addressing these issues, focusing on standardizing data formats, ensuring compliant data storage, and governing shared computational resources. In this context, the manuscript's approach may seem overly simplistic and somewhat naïve, especially in its assumption that readily available resources can singularly address these complex challenges.

The impact of adopting the proposed methods in radiology could initially seem substantial. However, there are potential limitations, such as researchers in certain Asian countries being unable to access tools like Google Colab at all, other countries (like all EU member states) typically not allowed to use it for many medical datasets, many institutions denying to provide access to their data via public cloud storage, etc. Moreover, the risk of Google making significant updates to their cloud's software stack, thereby disrupting established pipelines, is a critical concern that remains largely unaddressed in the manuscript but is likely to cause many of the collected pipelines to stop working properly. (Once this happens, who would fix them?).

In conclusion, while I appreciate the ad-hoc approach and the mindset of the authors, and acknowledge the need for cloud hosting and adherence to FAIR principles in our global community, my critique primarily centers on the authors' oversimplified perspective on resolving these multifaceted issues.

Reviewer #2 (Remarks to the Author):

The authors have addressed all my comments.

Point-by-Point Response to the Reviewers' Comments - SECOND ROUND

Reviewer #1

Remarks to the Author:

The revised manuscript exhibits significant improvements in response to the feedback from all three reviewers. The focus on the timely and important topic of reproducibility in computational pipelines is commendable and currently a subject of global discussion.

However, the manuscript has narrowed its scope to primarily address radiologists. This change, seemingly in response to the critiques raised in the reviews, comes across more as a superficial remedy rather than a comprehensive solution addressing the fundamental issues highlighted.

A notable concern, from my perspective, is the feasibility of implementing the proposed solutions for accessible and reproducible analysis pipelines across different regions. For instance, while the United States shows openness to using cloud providers for storing and processing scientific data, the stringent GDPR regulations in Europe and other data storage restrictions might impede the global applicability of the presented work.

The authors' intentions are certainly praiseworthy, and the subject matter is both timely and significant. However, the solutions presented seem rather improvised. While this does not inherently undermine their value, a more comprehensive solution would likely involve intricate global policies and political considerations. Current efforts involving governmental and scientific bodies are actively addressing these issues, focusing on standardizing data formats, ensuring compliant data storage, and governing shared computational resources. In this context, the manuscript's approach may seem overly simplistic and somewhat naïve, especially in its assumption that readily available resources can singularly address these complex challenges.

The impact of adopting the proposed methods in radiology could initially seem substantial. However, there are potential limitations, such as researchers in certain Asian countries being unable to access tools like Google Colab at all, other countries (like all EU member states) typically not allowed to use it for many medical datasets, many institutions denying to provide access to their data via public cloud storage, etc. Moreover, the risk of Google making significant updates to their cloud's software stack, thereby disrupting established pipelines, is a critical concern that remains largely unaddressed in the manuscript but is likely to cause many of the collected pipelines to stop working properly. (Once this happens, who would fix them?).

In conclusion, while I appreciate the ad-hoc approach and the mindset of the authors, and acknowledge the need for cloud hosting and adherence to FAIR principles in our global community, my critique primarily centers on the authors' oversimplified perspective on resolving these multifaceted issues.

We thank the editorial team and the reviewers for their help and feedback on our manuscript, and we appreciate the reviewers' recognition of the importance of our manuscript as well as the significant improvements we made.

Our work demonstrates the potential of cloud-based infrastructure for the sharing of transparent and reproducible AI-based pipelines (with radiology as a use case). Such resources help investigators interrogate each step of the complex AI pipeline (all the way from data retrieval to the reporting of the final results), and can thus accelerate the adoption of such pipelines by other researchers in the field and, ultimately, in the clinic. While all components of the AI-based pipelines need to be publicly available (which is often not a problem as many datasets are already publicly available, and AI code does not contain PHI and can easily be shared publicly), our framework can streamline the deployment of such pipelines in local environments, including private hospital systems.

Furthermore, as is apparent from the literature, the replication of AI research in radiology and, more broadly, in healthcare continues to be problematic, even for users with advanced technical skills. While we agree with the reviewer the proposed solution will not solve all problems, we believe a large number of scientists worldwide could already use these tools and greatly benefit from them today.

Regulations, such as the European GDPR, are indeed stringent but do not inhibit the use of our solution. What we propose does not force the authors to share anything new, and certainly, nothing that would fall under the purview of regulations such as the GDPR. Rather, we propose to 1) change the medium/infrastructure for sharing what they are already sharing and 2) make it easier to evaluate what they are already sharing on public images (which, by definition, are compliant with regulations such as GDPR). We designed our approach to be a foundational blueprint that serves as a starting point for adaptation to various constraints, e.g., by a third party, in order to comply with specific institutional restrictions. For instance, in the case authors want to use our approach using private data as suggested by the reviewer, they could do so in their own compliant environments (e.g., see “Using Cloud-based Resources with Local Runtimes” in the Supplementary Information). Following our proposal will result in having a reproducible and transparent starting point to ensure the AI models work as they are supposed to work, even if the data the pipeline is tested on in this hypothetical case will not be shared.

Furthermore, we believe the sharing of transparent and reproducible pipelines is complementary to and not mutually exclusive of the global efforts mentioned by the reviewer (i.e., the focus on standard data formats, compliance of data storage, and the governing of shared computational resources). Indeed, the unavailability of the cloud resources we use in our work in some countries can pose a problem with the proposed solution. Even though governance of data access and cloud computing access is outside of the scope of this manuscript, we recognize this is an important point, which we mentioned in the discussion of the revised manuscript.

We recognize the implementation and sharing of resources following the proposed workflow do not eliminate the need to maintain the code repository over time, and we highlighted this limitation in the revised discussion section. We believe the constantly evolving field of AI makes sharing something that is time-proof harder than in other fields of computational research. However, we find that the addition we made to the Supplementary Information (i.e., “Using Cloud-based Resources with Local Runtimes”) can at least partially limit the disruption that significant updates to the cloud-based computational components could have on the pipelines (since the Docker images powering Google Colab are publicly shared and can be run, in different versions, at any point in time). Moreover, we believe, as we discussed in the Supplementary Information, this makes the transition between local and cloud processing seamless - and therefore opens the possibility of using the proposed workflow on data that cannot leave the premise (such as the example the reviewer pointed out regarding the limitations in the sharing of medical data within the EU).

Finally, we would like to note that we agreed to change the title of the manuscript as suggested by the reviewers, not as a superficial remedy to some of these issues - but rather to make the topic of our work clearer and better, directing the manuscript toward its intended audience.

We have revised the manuscript and have expanded the discussion section to capture the complexity of the topic better and highlight some of the limitations pointed out by this comment. In particular, we added the highlighted paragraph to the Discussion to address the reviewer’s concerns regarding the limitations on the access to cloud resources from specific geographic regions, and the governance of data access (the paragraph before is reported for context):

[...] However, it must be recognized that using the cloud for data sharing, particularly for reproducibility purposes, involves additional steps. For instance, anonymization and institutional approval are often necessary to address security and privacy concerns about sharing protected health information, such as medical images and associated clinical data. In the cases when, for the aforementioned reasons, sharing data is impractical, the users might

prefer to opt for techniques such as federated learning, as it enables collaborators to build and refine AI models without sharing raw data (only model updates are exchanged, ensuring that sensitive patient information remains localized and secure). Furthermore, the proposed solution assumes that the data needed for the notebook and the cloud-computing platform itself are available in the region in which the user is trying to reproduce the experiments. Governance of the data access and the cloud-computing access is outside of the scope of this manuscript, but it is important to recognize this can pose a limitation to our workflow.

Furthermore, we expanded the discussion to acknowledge the other remark from the reviewer on the maintenance of such pipelines (the paragraph before is reported for context):

Furthermore, cloud providers might update these components without appropriately informing the users, resulting in the disruption, or even failure, of the previously developed pipelines. Even if this does not entirely hinder the benefits of the added transparency, it is essential to note the implementation and sharing of resources following the proposed workflow and through the aforementioned cloud services do not eliminate the need to maintain the code repository over time. Finally, cloud computing may not always be the most cost-efficient solution for all use cases, particularly for long-term or high-usage applications.